# Hopeful, but Worried for the Future: An Analysis of the Lived Consequences of Colonisation as Narrated by Older South Sámi in Norway

**Tove Mentsen Ness** [1,*] **and Mai Camilla Munkejord** [2]

1  Centre for Sámi and Indigenous Studies, Faculty of Education and Art, Nord University, Høgskolevegen 27, 7600 Levanger, Norway
2  Centre for Care Research West, Western Norway University of Applied Sciences (HVL), Årstadvegen 17, 5009 Bergen, Norway; mcmu@hvl.no
*  Correspondence: tove.m.ness@nord.no

**Abstract:** Various forms of colonisation and discrimination processes are unfortunately common to Indigenous Peoples worldwide. In this article, the focus is the lived consequences of colonisation in the Norwegian part of Sápmi (the Sámi traditional lands), where systematic state-imposed colonisation officially ended decades ago. Thus, based on a thematic analysis of qualitative in-depth interviews with 12 South Sámi aged 67–84, the aim was to examine how stories about hopes and worries for the future can shed light on how colonisation is experienced among older South Sámi today. The voices of these participants are important, as they can be considered triply muted, due to (a) ageism and (b) continued yet unconscious colonising practices against the Sámi in general and (c) against the South Sámi people in particular, a minority within the minority. Inspired by decolonising perspectives, this article reveals that older South Sámi are worried for the future, not only due to memories from the past but also their experiences with persisting colonial practices such as the ongoing enlargement of windmill parks and cabin areas in the midst of the winter pastures of the South Sámi reindeer herders. Hopefully, the time has come to finally put an end to colonial practices and take collective responsibility for creating a more just future for both coloniser and colonised. Similarly to the participants in this study, the report from the Truth and Reconciliation Commission also stresses issues such as loss of language, experience of racism, and reindeer husbandry being under pressure. This report may therefore be used as an important tool to ameliorate the conditions of the Sámi people if taken into consideration in the time to come.

**Keywords:** indigenous peoples; South Sámi; Norway; older people; lived experience of colonisation; collective responsibility for the past; decolonising perspective

## 1. Prologue

In 2010, after years of turmoil, Fosen Vind AS was granted a licence by the Norwegian Water Resources and Energy Directorate to establish a windmill park in a mountainous area of central Norway, in the middle of the South Sámi land. As a result, Fosen Vind AS constructed one of the largest land-based wind power plants in Europe, with six wind farms in this area [1], while the land of South Sámi reindeer herders, who for generations had collectively used the area for winter pasture, was expropriated. These decisions were later confirmed by the Ministry of Petroleum and Energy. The South Sámi people have since then persisted in their fight for the right to continue reindeer herding in this territory. But in 2020, the windmill park was completed, becoming the largest windmill park on land in Europe.

One year later, however, in October 2021, the Norwegian Supreme Court argued that Fosen Vind could have constructed the wind plant on another piece of land that would have caused less harm to the reindeer husbandry in the region. Thus, referring

to the United Nations Declaration on the Rights of Indigenous Peoples (UNDRIP), the Supreme Court concluded that the establishment of the windmill park was a violation of the rights of the South Sámi people to maintain, control, protect, and develop their cultural heritage. According to lawyers Andreas Brønner and Eirik Brønner for the South Sámi reindeer herders, this verdict entailed that the licence, the expropriation of the land, and the windmill park itself were to be considered illegal, and they expected the windmills to be removed [2,3].

## 2. Introduction

Officially, the colonisation process in Norway ended in 1980 with the so-called Alta controversy, when Sámi and allied activists were able to reduce the size of a planned hydro-electric power dam that would have put a Sámi village called Masi and surrounding grazing lands under water [4]. In 1968, the first plans for development of the Alta-Kautokeino watercourse was presented, entailing that Masi and surrounding grazing areas would have been under water. The plans met strong resistance from the Sámi reindeer organisations and environmental activists. After years of political struggle, in 1973, Masi was exempted from the construction plans, but the rest of the plans were approved by the Norwegians, and the Alta watercourse was dammed [5]. This controversy, later called the Alta controversy, represented a political shift in a more Indigenous-friendly direction, which was confirmed by the adoption of the Sámi Act in 1987 and by the establishment of the Sámi Parliament in 1989, which is a democratic structure representing the Sámi people vis à vis the Norwegian state. Moreover, in 1990, Norway ratified ILO Convention No. 169, called The Indigenous and Tribal Peoples Convention, and thereby officially recognised the Sámi as an Indigenous People [6]. From 2023, the Constitution of Norway (§ 108) also stresses that the Sámi, as an Indigenous population, have the right to their own language, culture, and social life. In practice, however, the establishment of the Fosen windmill park may be considered but one example of colonial practices that continue to take place in various parts of Sápmi even today.

In this article, the focus is on the lived consequences of colonisation in a country where systematic state-imposed colonisation has officially ended [4], thus highlighting the persistence of such processes in the everyday lives of the people concerned.

Inspired by decolonising perspectives, as outlined below, this article aims to raise the collective consciousness regarding the situation of the Sámi people in Norway. This is done by highlighting the experiences of older South Sámi, whose voices can be seen as triply muted due to (a) ageism and (b) continued yet unconscious colonizing practices against the Sámi in general and (c) against the South Sámi people in particular, a minority within the minority, as they only number around 2000 people living in the Norwegian and Swedish parts of Sápmi [7] of a total population of 80,000–95,000 [8]. This article reveals that, while the participants are hopeful, they are deeply worried for the future of their people. Their worries are not only caused by memories from the past but also based on their experiences with persisting colonial practices such as the ongoing enlargement of windmill parks and cabin areas in the midst of the winter pastures of the South Sámi reindeer herders. In conclusion, in line with Mackinlay and Barney [9], we argue that everyone in society, but especially all those who represent the majority population, has to take collective responsibility for the past in order to heal the wounds of both the colonised and the colonisers with the aim of putting an end to ongoing unacceptable practices.

### 2.1. Background

The Sámi people live in the northern parts of Norway, Sweden, Finland, and Russia [8], and their homeland Sápmi extends across those four countries. Traditionally, the Sámi people lived both along the coast, where they combined fishing and small-scale farming, and in inland areas, where they lived as reindeer herders maintaining a mobile lifestyle. In the Norwegian part of the South Sámi territory, there are 16 reindeer grazing districts with the equivalent of 150 full-time jobs [10]. However, because it is common to work part-time

in reindeer herding, substantially more than 150 South Sámi are involved in the industry. Not only the individual reindeer owners but also the broader community, called the *sijte*, are involved in taking care of the animals [11]. The *sijte* varies in size throughout the year and consists of people of all ages, including children and older women and men [11].

The use of the South Sámi land is intimately tied to how the reindeer herds move across the landscapes, which may vary considerably according to season and weather conditions. Section 4 of the Reindeer Herding Act states that the South Sámi population in the South Sámi area has the right to practise reindeer herding in central Norway based on immemorial use. This right has been confirmed repeatedly by legislation and court rulings since the 19th century [12]. The needs of the reindeer in the South Sámi territory, however, are increasingly being threatened by the establishment of cabin areas and windmill parks [13]. This causes worries, but also ignites sparks of hope that a change will come, as will be illustrated in this article.

The Norwegianisation process started in the late 1860s and was strengthened with a regulation from 1880 stating that all Sámi and Kven (Norwegian Finns) children in the so-called transitional (multi-ethnic) districts in northern Norway had to learn to speak, read, and write the Norwegian language [4]. Later, in 1898, a school decree was issued stating that the Sámi and Kven languages had to be limited to "strictly necessary" use, and even required teachers to prevent non-Norwegian pupils from speaking their mother tongue during breaks. This policy was practiced in all parts of northern Norway, perhaps most systematically in boarding schools [4]. Thus, several testimonies cited by Minde [4] indicate that Sámi pupils in various parts of Sápmi over many decades were victims of derogatory attitudes from teachers, including systematic neglect and bullying. One consequence of the Norwegianisation process was that many Sámi parents from the early 20th century avoided speaking their mother tongue to their children in order to save them from the shame of being "culturally inferior" to the majority population [4,14–16]. As a result, it is estimated that today only a few coastal Sámi, and only 25% (500–600 persons) of the South Sámi population, speak their native language [17,18], and many still deny, or are not even aware of, their Sámi past [4].

Havika Boarding School was established in Norwegian Sápmi by the Sámi People's Mission in 1910 and is a symbol of both the Norwegianisation and Christianisation of South Sámi society [19], even though Risto [20] highlights that the South Sámi attending the boarding school had varying experiences and that some teachers did allow the children to speak their mother tongue among themselves. The assimilation process was gradually abandoned with the establishment of a boarding school in Hattfjelldalen in 1951 and the Sámi School in Snåsa in 1968 [21]. However, the South Sámi language was not officially recognised as equal to Norwegian until 2008, when Snåsa municipality, as the first municipality in the South Sámi area of Norway, was incorporated into the Sámi administrative area [22].

## 2.2. Theoretical Perspectives on Colonisation

Various forms of colonialisation and discrimination processes are unfortunately common to Indigenous Peoples worldwide. Colonisation can be defined in this context as different forms of assimilation processes that have taken place in conjunction with the establishment of nation states in most parts of the world. Examples are the Norwegianisation of the Sámi referred to above [4], the Germanisation of minorities in Germany [23], and Sinicisation, which relates to the imposing of Chinese language, values, and worldviews onto ethnic Indigenous Peoples and other national minorities in various parts of Asia, such as in Taiwan [24]. While colonisation in North America entailed a relatively short period of invaders coming from outside to at least partially exterminate the native peoples and appropriate their lands, in Norwegian Sápmi, where this study took place, to cite Skille [25], colonisation has been more about "defining right and wrong, morally and legally, when it comes to artefacts and behaviour as well as values and ontology". In other words, according to Skille [25], colonisation in the Nordic context may be understood as imposing

on others one's "definition of rights regarding worldview, knowledge creation and inter-human behaviour". Skille [25] also shows how government authorities exploited science to suppress the Sámi population in Sámi–Norwegian history, and therefore state assimilation policy was rationalised by "scientific evidence" that "proved" Sámi subordination. Before colonisers enforced their worldviews by means of governmental structures and various economic, educational, and legal measures, Indigenous Peoples had their own cultures, languages, cosmologies, and religions [26,27].

According to Mackinlay and Barney [9], the process of *decolonisation* emphasises a moral need to rectify the mistakes of the colonial past in order to foster social justice for all involved. Inspired by these insights and by an understanding of decolonial practice as the challenging and undoing of colonial logics and practices, we explore the lived consequences of colonisation among older South Sámi as a *decolonial practice*. Therefore, in this article, the aim is to examine how stories about hopes and worries for the future can shed light on how colonisation is experienced by older South Sámi today.

### 3. Methods

*3.1. Design and Choice of Methods*

In this article, a qualitative research design was used to explore the lived experiences of colonisation through an analysis of how older South Sámi envision the future. Individual narrative interviews were conducted [28]. Patton [29] stresses that narratives can reveal and communicate our human experiences, our social structures, and how we as humans make sense of the world.

*3.2. Participants*

Twelve older South Sámi, seven men and five women, living in the South Sámi area of Norwegian Sápmi, agreed to participate in this study. At the time of the interview, they were aged from 67 to 84 years, with a mean age of 74 years. While some of them had been active in reindeer husbandry their whole lives, others had not actively taken part in this lifestyle since they were young. All participants lived in their own homes. Some, however, had a semi-nomadic lifestyle, commuting between at least two different homes in order to follow their reindeer, which lived in one area in summertime and in another area in wintertime. The participants in this study were not asked if they were fluent in the South Sámi language, but most participants mentioned if, and how, they used or had used the language themselves. Some of them revealed that they used South Sámi on a daily basis as their main language, others said that they sometimes spoke some South Sámi, and some had little knowledge of the language, either because they had forgotten it or because they never learned it, even as children.

*3.3. Recruitment and Data Collection*

To gain access to the field, Tove Mentsen Ness contacted two acquaintances in the South Sámi community who provided names and phone numbers of potential participants. In addition, snowballing was used [30]. This means that some of the participants suggested other candidates who might have wanted to take part in the study. Six participants were recruited through the two acquaintances in the South Sámi community, and the other six were recruited via suggestions from the participants.

In the interviews, the participants were invited to share their life stories. Portelli [31] emphasises how telling one's story is an art form, and a cultural and personal meeting between the storyteller and the listener—in this case, the participants and the interviewer (Tove Mentsen Ness) of this study. The participants talked about their childhood, the family they grew up in, their education, their professional experience from reindeer husbandry and other work, and their own family. Of particular interest for this article, the participants were also invited to shed light on their hopes and worries for the future, for themselves, for their families, and for the Sámi community as a whole. Even though Tove Mentsen Ness does not have an Indigenous background, she has conducted several studies in different

parts of Sápmi and has developed an extensive personal network, especially in the South Sámi community, over many years.

Inclusion criteria for taking part in this study were having a South Sámi background and being over 65 years of age. These criteria were mentioned during the recruiting process. All participants self-defined as Sámi. The interviews were conducted, recorded, and transcribed verbatim by Tove Mentsen Ness in Norwegian, as she does not speak South Sámi. In fact, several of the participants in this study had lost or never learned South Sámi as a child.

The interviews were conducted between August 2019 and February 2020. A total of 24 interviews were conducted (two with each participant), all of which took place in the participant's own home, except for one participant who wished to conduct the interview in a café. The interviews lasted from 47 min to 3 h 23 min (mean: 2 h 4 min).

### 3.4. Data Analysis

When analysing the data, a reflexive thematic approach was used [32,33]. In the following, we (the authors) explain how the analysis was performed to enable the reader to assess the trustworthiness and relevance of the findings, as stressed by Stige et al. [34]. In the first phase, the authors read the transcripts several times individually to achieve familiarisation with the data set as a whole. In this phase, the authors made preliminary notes and identified preliminary themes. Second, each of the authors used an inductive, bottom-up approach to code the material. During the coding phase, the authors met digitally on Teams in order to discuss and reflect on preliminary codes. After this phase, the authors constructed themes, where the previous coding formed the basis. In this phase, the themes were developed and later revised in relation to the research topic posed [33]. Consensus was reached after several meetings and conversations among the authors, and the names of the themes were refined. Finally, the authors decided to use a decolonising perspective to shed light on the findings in light of the research topic.

### 3.5. Ethical Reflections

Some of the literature on Indigenous methodologies suggests that Indigenous research should only be conducted by Indigenous researchers; see an analysis of this by Olsen [35]. While we wholeheartedly support the argument that positivist and imperialist research has caused much pain among Indigenous Peoples worldwide, we believe that it is important that the field of Indigenous studies include researchers with mixed ethnic background and non-Indigenous background, who can be seen as "allied others". The authors of this article define themselves as allied others aiming to contribute to promoting justice and well-being for Indigenous Peoples. Moreover, they decentre themselves to be able to understand the standpoint of "the other". The participants are always paramount, and the authors believe that this is a particularly important exercise for non-Indigenous researchers conducting research with and among Indigenous Peoples, because this process necessarily has "historical roots with political and cultural implications" [25].

It should be noted the authors of this study have chosen not to specify whether the participants are female or male. This is to ensure anonymity because the South Sámi population is very small. This decision is in line with Surmiak [36], who stresses the importance of total anonymity in research on vulnerable populations, in particular when people may easily be recognised, which is the case for the participants of this study (older South Sámi from reindeer-herding families). Because of this, the authors of this article choose to refer to all participants as he/him, and their family members or other persons they talked about as she/her. It should be noted that the aim of this study was to analyse the participants' hopes and worries about the future, not possible differences between the male and the female participants, and whether this affected their experiences.

*3.6. Ethical Considerations and Roles on the Research Team*

Before taking part in this study, informed written consent was obtained from the participants. In addition, they were informed about the possibility to withdraw at any time with no explanation required. They were promised confidentiality in publications, and the authors have therefore anonymised the quotes and examples used in the Results section of this article. Further, the study was carried out in accordance with the Declaration of Helsinki [37] and was assessed by the Norwegian Agency for Shared Services in Education and Research (SIKT) (No. 788125). Both authors are privileged researchers. Tove Mentsen Ness has worked for several years as a registered nurse and teacher in a rural setting in the South Sámi area and conducted several studies in this area, while Mai Camilla Munkejord has extensive experience as a social anthropologist who has conducted fieldwork in various parts of Sápmi over the years. The analysis was conducted from an outsider (non-Indigenous) position, and was therefore influenced by the authors' experience and background. This is described as an empowered and privileged position by Olsen [38]. Both authors contributed equally to the planning of this study, the analysis of the data, and the writing and revision of this article, and are therefore named in alphabetical order.

## 4. Results

The analysis revealed three themes: "hopes and worries related to racism", "hopes and worries about the South Sámi language", and "hopes and worries for the future of reindeer husbandry", as presented below.

First and foremost, the participants hoped that the Sámi people, as a socio-cultural group with their own values and lifestyle, would survive. Despite their own sometimes challenging life experiences, the participants were hopeful for the future because they observed that the younger generations were increasingly making efforts to revitalise the South Sámi language, engage in reindeer husbandry, and stand up to unfair treatment from the majority population. In the following, these findings will be elaborated.

*4.1. Hopes and Worries Related to Racism*

Some participants recalled how, in their childhood and youth, they were sworn at and called derogatory names such as *finn*. *Finn* is a word that was previously used to refer to Sámi, often in a derogatory way. One of the participants, who was called all sorts of shaming names when he was young, said:

> "So, I thought... Like I was almost an adult before I discovered that it's not common to be called all sorts of things. That not all people are sworn at, that this kind of thing is not normal in everybody's lives. So... It worries me. I'm not talking about myself, but for my... I really think that in general, there are dark clouds hanging over the Sámi people."

Related to this, several of the participants stated that they hoped that their family members would be accepted for being Sámi and that they would not face the same prejudice and bullying as some of them had faced, especially when they were young, but even today. This was because, as a Sámi family, and in particular if the family was engaged in reindeer husbandry, they were often exposed to bullying or racism from the majority population, even now. Some shared that their children or children-in-law sometimes had to cope with ethnically based criticism or even racism for being a reindeer herder. One of them gave this example when talking about the coming generations:

> "Of course, I hope that everything will go well with them [children and grand-children] And that they will be accepted for being Sámi. That they are allowed to be Sámi, and that they don't have to hide, as our generation and those before us had to do. But we see how some of them [the younger ones] must struggle even today, and that's awful to watch. Yes, it is."

Some of the participants have experienced racism from time to time, even today, which was hard for them to bear. The following story from one of them may illustrate this. This

participant said that he was waiting by his car outside a shop while his wife was buying ice cream. While he was standing there, a couple of children came towards him. He did not know them but estimated them to be around 10–12 years of age. They stopped by the car and asked him if he was a relative of a deceased person called Emma. He confirmed that she was indeed a close relative, for a moment believing that these two children had approached him to share some compassion, but this was not the case. Their next move was very surprising and unexpected for the participant, when one of the two children said:

> "You should have been dead too, so we could have got rid of some more 'finns'"
> [local derogatory slang for Sámi].

The participant continued his story, commenting to the researcher:

> Participant: "I was so surprised. But I asked them: Who says that? Well, they said, it's mum and dad! I still regret. . . I didn't manage to think. . . I should have asked them their names, but I didn't. So you know it's hard sometimes. (. . .) I don't think it was the children [who came up with this xenophobic idea]. I think they'd heard this at home. I think it's the parents that talk this way. It frightens me."

> I: "Yes, that must have been unexpected."

> P: "Yes, it was. So I think... You must be strong in your mind to manage to survive in the society we live in."

In line with this, some mentioned that the Truth and Reconciliation Commission had recently been established in Norway in order to listen to the colonial harms of the past. However, some doubted that their voices would be truly heard, especially because the wrongdoings of the past, according to them, had not yet ended:

> "They talk so much about the Truth and Reconciliation Commission and what was done to the Sámi people *before*. But it is more or less the same way today, really."

Some participants said that they hoped that in the future the Sámi would be heard and that they would be treated well and not exposed to racism. Indeed, several of them were quite hopeful as they observed that the younger generations of Sámi stood up for themselves and demanded equal treatment on terms with the majority Norwegians. One of them said:

> "It seems like the younger generation has woken up. They don't tolerate that they [majority Norwegians] say or write this and that. The younger generations, they stand up and speak up! That is what we have waited too long to do."

### 4.2. Hopes and Worries about the South Sámi Language

As mentioned in the introduction, only about 25% of the South Sámi people are estimated to speak the South Sámi language. Consequently, several of the participants of this study were concerned about the future of the language. While not all of them spoke South Sámi today, most of them had learned it as their first language but no longer spoke it in their daily lives. Several, however, mentioned that their adult children and grandchildren currently made efforts to learn South Sámi by taking language classes if they could not already speak it. This made them hopeful that the coming generations would use the South Sámi language more than they had. One participant, referring to the younger generations, said:

> "Several are very active in using the South Sámi language nowadays, and some of them take courses to learn to speak the language."

Some of the participants had learned South Sámi as their first language, but had forgotten the language, for instance, in schools where their mother tongue was forbidden. These participants were concerned about what would happen if they got dementia and went back to their first language. Several had heard stories of that happening. For those who had adult children who spoke the language or were learning to speak it, this future scenario represented security, and one participant stated:

"Yes, I am a little afraid of going back to my first language, but my son is learning South Sámi now. That makes me feel safer."

Others, however, worried that if they lost the Norwegian language in favour of their first language, they might not have anyone to talk to because neither their wife nor their children knew South Sámi. One of them noted that if he spoke Sámi after moving to a nursing home one day in the future, the healthcare staff might have difficulty understanding him. He said:

"Perhaps they will not even understand that I am speaking South Sámi, and I find that a bit worrying. Maybe they'll think I'm just 'babbling', not realising that it is a language?"

This participant was somewhat worried to think about possibly speaking South Sámi later in life, as he had married a non-Indigenous woman, and thus neither his wife nor their children could speak the language. He hoped that other persons in his family would be able to translate for the healthcare staff, if necessary, as other family members had used South Sámi to a much higher degree than he had.

*4.3. Hopes and Worries for the Future of Reindeer Husbandry*

The long struggle against the establishment of the Fosen windmill park in the mountainous part of the South Sámi area in Norwegian Sápmi, as referred to in the introduction, had caused substantial worries about the future of reindeer husbandry, not only for the South Sámi reindeer herders directly involved but also for the Sámi community as a whole. Thus, when the participants were asked about their hopes or worries for the future, most of them talked at length about whether reindeer herding would be allowed to survive at all. Threats in this regard were not only related to the loss of land due to the windmill park and the establishment of new cabin areas in the mountains but also to the associated infrastructure, roads, movement, and noise, all of which disturb the reindeer. Some of the participants worried that they would either have to reduce the size of their herd or be forced to find new ways to manage or feed their animals. One participant said:

"It is of course possible to reduce the herd or to substitute the traditionally used winter grazing areas by feeding them [by giving supplementary feeding], but personally I think that's completely wrong."

Thus, several participants said that if the situation continued in the same vein as for the past couple of decades, with ever increasing windmill parks and cabin areas in their territory, in addition to the existing challenges related to natural predators such as wolves and eagles, they would end up being forced to quit reindeer husbandry altogether. One said:

"You know, our lands are being expropriated. That is very negative for us [the reindeer herders]."

And he continued:

"It started with the alteration of the big lakes. That led to a loss of big grazing areas and heathlands due to reduced areas, but also the consequences of the alteration [the reindeer shun previous grazing areas, e.g., because of the sound from the windmills]. And now we have the big propellers [windmills] all over the place."

The worries for the future of reindeer herding were also related to concerns about the expansion of cabin areas with their associated infrastructure and about ongoing climate change. Some noted that in the future it could be more difficult for the reindeer to find lichen and other foods in the mountainous winter pastures due to less snow and more ice in the winter. One of the participants expressed his concern in the following way:

> "I'm not concerned yet, because we tend to find solutions, and we have always adapted to changes. But finding enough food [in the mountains] will be difficult if the winters continue to be like this one."

Despite worries of different kinds and along different lines, several were optimistic for the future of reindeer husbandry. One of the participants said:

> "I am very interested in the future for reindeer husbandry because I have children and grandchildren in reindeer husbandry (. . .), so I really hope there is a future for them there. And personally, I think there is a future for them in reindeer husbandry, I do."

He explained that the reindeer herders were strong and accustomed to having to struggle. He himself had been faced with the consequences of the Chernobyl accident, which caused polluted rain and led to challenging situations for reindeer husbandry. He said that, over the years:

> "So much has gone against reindeer husbandry. But reindeer husbandry is strong."

Several participants hoped that reindeer husbandry had a future, because they had survived through challenging times before. However, as one stated at the end of one interview, it meant that more people, particularly decision-makers, in broader Norwegian society had to realise that reindeer husbandry needed to keep its grazing areas. Specifically, in relation to the ongoing Fosen case, he said:

> "Society needs to accept that the reindeer need the grazing areas in order to survive. Then, and only then, can reindeer herding survive in the future, if not it will simply not survive."

## 5. Discussion

In this study, the aim was to examine how stories about hopes and worries for the future can shed light on how colonisation is experienced by older South Sámi today. The analysis revealed that the participants had both hopes and worries with regard to racism, the South Sámi language, and reindeer husbandry. They hoped that future generations would not have to experience racism or discrimination in the same way as they themselves. While some of them experienced racism or systematic bullying when growing up, some have also experienced discrimination today. Being subject to racism and discrimination because of their Sámi identity is not exclusive to the participants in this study; it has been identified in several other studies of Sámi people in Norway, e.g., Hansen [39] and Hansen and Sørlie [40]. Moreover, Omma, Holmgren, and Jacobsson [41] found that experiences of racism and discrimination are most prevalent among Sámi practicing reindeer herding as a lifestyle. Despite these experiences, some participants hoped that the younger generations of Sámi would not suffer from discrimination in the same way as they themselves because the younger generations seemed to be more resilient and outspoken, and therefore would hopefully be able to re-establish justice for the Sámi people.

The participants were concerned about the future of the South Sámi language. They also worried about which language they would be able to speak themselves if they were to develop dementia, and were unsure whether they would be able to make themselves understood if they went back to their mother tongue. Primarily, however, they were concerned about whether the South Sámi language would survive. Several of them were quite hopeful, however, as younger family members who did not speak South Sámi from childhood had started to learn the South Sámi language through courses and classes. The Sámi language can be seen as a bearer of culture [42] and part of one's identity as a Sámi [43]. Using one's mother tongue may be essential when needing to express oneself in certain situations. This is confirmed in a study by Mehus, Bongo, and Moffitt [44], where Sámi patients speaking a Sámi language often preferred to use their mother tongue to be sure to make themselves understood and thus to feel safe. Since many South Sámi are no longer able to speak the South Sámi language, questions about the future of this language were a

common theme among the participants. However, some of them stated that although they could no longer speak South Sámi themselves, they felt that future generations might learn and thus revitalise the language of the South Sámi people.

The participants were also concerned about the future of reindeer husbandry. They were worried about the expansion of windmill parks and cabin areas in their winter pasture lands and the negative effect this would have on the possibility to continue reindeer husbandry in the same way as in previous years. However, they hoped, and most of them also believed, that reindeer husbandry would survive as a lifestyle for future generations. These concerns are echoed in previous studies, such as in relation to the loss of grazing land due to industrialisation or so-called modernisation processes and to climate change [45]. According to Risvoll and Hovelsrud [46], loss of land is seen as the biggest threat to the future of reindeer herding. It is important to emphasise that the reindeer in the South Sámi area are semi-wild, and to avoid stress they need to stay away from infrastructure such as cabins, roads, and windmills [47]. This is especially the case in vulnerable periods such as during calving and when the calves are young [13]. Brown, Flemsæter, and Rønningen [13] also identified concerns related to carnivores. The participants in this study thought that the loss of land could force them to conduct reindeer herding differently than before, such as giving supplementary fodder, which may lead to the reindeer becoming domesticated livestock [48]. Despite all of these concerns and worries, the participants hoped that reindeer herding would survive as a lifestyle for future generations. These findings emphasise the importance of reindeer herding to the Sámi community. It is not only a matter of regularly having to fight against competing interests and companies in order to secure access to grazing lands. More than that, reindeer herding constitutes what we may call a material foundation for Sámi culture, such as food, clothing, language, and handicraft, and it therefore also carries a very important symbolic meaning [49].

The colonialisation of the Sámi was never a military invasion but evolved as a socio-cultural and political assimilation or Norwegianisation process that over time defined "right and wrong" morally, legally, culturally, spiritually, and linguistically (e.g., [25]). However, while Norwegianisation officially ended in Norway several decades ago [4], this article indicates that the consequences of past colonisation still shape everyday experiences and hopes and worries for the future among older South Sámi today. As a society, we have a collective responsibility for making sure that past wrongdoings are openly addressed in order to heal the wounds of the past [9].

In this regard, the Truth and Reconciliation Commission was established in Norway in 2018 to lay the foundation for recognition of the consequences of the assimilation processes of the Sámi, Kvens/Norwegian Finns, and Forest Finns. The mandate of this commission was to conduct a historical survey of Norwegian authorities' wrongdoings, to study the effects of the Norwegianisation policy, and to suggest measures to lead to further reconciliation (Innst. 408S 2017–2018). It was not entirely clear whether the Truth and Reconciliation Commission also would shed light on ongoing discrimination against Sámi, Kvens/Norwegian Finns, and Forest Finns or whether it is taken for granted that such discrimination no longer exists. The Truth and Reconciliation Commission report was published on 1 June 2023. It described the injustices perpetrated on the Sámi populations for centuries [49]. One of the main findings in the report is that the Norwegianisation process has led to the loss of the Sámi language for many Sámi, which also applies to the South Sámi language, with the result that this language is severely threatened. This is in line with the worries of the participants in this study, as several participants highlighted that they had lost the Sámi language or never learned Sámi themselves. Even though the participants had hopes for future revitalisation of South Sámi, as several of their children were learning the language, whether the language will survive is still an open question.

Moreover, the commission report highlights that racism and negative attitudes towards the Sámi people have existed for generations and are still present today. This also concurs with the views of the participants in this study. The commission report highlights that one reason for this is that the majority population has too little knowledge about the

history and background of the Sámi population. Another finding in the report is that the educational system over several generations has contributed to the loss of the Sámi language; therefore, it must become an arena for the revitalization of the Sámi language and culture going forward. This finding is also in line with the participants' experiences, as several highlighted that they lost their Sámi language when they were in boarding schools.

Another main finding in the commission's report is that reindeer husbandry has been strongly affected by the Norwegianisation process. This has led to loss of grazing areas, limited possibility to transfer traditional knowledge to children and grandchildren, as well as threatening the Sámi language. The participants of this study expressed similar concerns. The Fosen case is specifically mentioned in the commission's report, which states that the consequence of this case has not only been the loss of important grazing lands but also that this case in itself has led to an increased threat to the South Sámi language and cultural knowledge of reindeer husbandry [49].

Reindeer husbandry is the pillar of South Sámi language and culture and is therefore essential to maintaining and preserving the language and culture [50]. This way of life should therefore be afforded protection. Thus, although the commission report mainly focuses on the injustices of the past, it also emphasises that the colonisation processes of the past are persisting under the radar, and that, as a result, discrimination of the Sámi population is still ongoing [49].

In line with the commission report, this article also indicates that discrimination and experiences of injustice are not only a matter of the past. Rather, as argued by the participants, the Sámi population still carries stories waiting to be told in order to re-establish justice, tolerance, and respect among colonisers and colonised. The Truth and Reconciliation Commission may therefore play an important role in this regard in the time to come. However, to enable this to happen, the overall policies, legislation, and guidelines have to change to make sure that the Sámi people have the possibility to preserve, develop, and learn about Sámi language and culture. Sámi language and cultural knowledge must become more visible in society as a whole, for example, in healthcare and education. In healthcare services, this will become more prevalent when the new health education curricula from 2019, which include mandatory learning outcomes about Sámi issues for all students, are more extensively implemented. This will eventually enhance knowledge of Sámi cultures, languages, and ways of life [51–53]. This also applies to the educational system in Norway, where there is a process of incorporation and recognition of the rights of the Sámi [54] and knowledge of Sámi language and culture. Additionally, this means that the majority population in Norway must increase its knowledge to stop the ongoing racism and negative attitudes. Awareness about the Sámi is needed on all levels of society—now.

## 6. Concluding Remarks

While colonisation in Norway officially ended around 1980 with the Alta dam controversy that became a symbol of the Sámi struggle against cultural and linguistic assimilation policies and practices, this article indicates that the consequences of colonisation still shape the everyday experiences, hopes, and worries for the future of older South Sámi. This is not only due to memories of various forms of colonisation in the past, but also the experience of persisting colonial practices such as the ongoing expansion of windmill parks and cabin areas in the winter pastures of the South Sámi reindeer herders, as mentioned in the introduction. However, the verdict from the Norwegian Supreme Court illustrates that we may hope that justice will increasingly become established in Norwegian society. The participants in this study remind us that the Sámi people in many contexts have been badly treated by the state and by representatives of the majority population over generations. More specifically, they express the hope that racism will come to an end, that reindeer herding as an industry and cultural lifestyle will survive, and that the South Sámi language will gradually be revitalised. In particular, they hope that the younger generations of Sámi will be brave enough to speak up and shed light on both past and present injustices. The report from the Truth and Reconciliation Commission on Norwegian Sápmi stresses the same

issues of loss of language, experiences of racism and consequences for reindeer husbandry as the participants in this study. The report may therefore be used as an important tool to enable change for the Sámi people if it is taken into account and used to promote Sámi language and culture in society as a whole. The future of not only the South Sámi but also of Indigenous Peoples all over the world is a matter of concern for all of us. It is time for change.

**Author Contributions:** Conceptualization, T.M.N. and M.C.M.; methodology, T.M.N.; validation, comparison of preliminary analysis, T.M.N. and M.C.M.; formal analysis, T.M.N. and M.C.M.; data collection, T.M.N.; writing—original draft preparation, T.M.N. and M.C.M.; writing—review and editing, T.M.N. and M.C.M.; funding acquisition, T.M.N. and M.C.M. All authors have read and agreed to the published version of the manuscript.

**Funding:** This project was funded by the Norwegian Research Council (grant number 287301).

**Institutional Review Board Statement:** The study was conducted in accordance with the Declaration of Helsinki, and approved by Norwegian Agency for Shared Services in Education and Research (SIKT), former NSD, Norwegian Centre for research data, and provided project number 788125.

**Informed Consent Statement:** Informed consent was obtained from all participants.

**Data Availability Statement:** Data in Norwegian language. Not available for international readers.

**Acknowledgments:** We wish to thank the participants for taking part in this study. Thanks also to our colleagues Grete Mehus and Jan-Erik Henriksen, UiT the Artic University of Norway, Wasiq Silan, the National Dong Hwa University, Taiwan and Pertice Moffitt, Aurora Collage, Canada for their valuable comments to an earlier draft of this article.

**Conflicts of Interest:** The authors declare no conflicts of interest.

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
