# Peer review of "Hopeful, but Worried for the Future: An Analysis of the Lived Consequences of Colonisation as Narrated by Older South Sámi in Norway"

_societies, doi:10.3390/soc14050071_

Round 1
Reviewer 1 Report
Comments and Suggestions for Authors
This article has a lot of potential and could be very good. However, unfortunately, in its present,t structure, I feel that it is greatly lacking in areas, organisation, and presentation.
I hope that the following suggestions will be of some assistance, and my intention is to draw attention to areas that need improvement rather than criticising it.
I think that the article is too short, and the conclusion is only a few lines. It needs far greater development.
The interviews are good sources to use to analyse, but they are badly presented. They are not distinguishable at all from the wording of the author(s). They are rarely in inverted commas, and they are not in any way distinguishable. At one point later on in the article, the author(s) introduce single quotation marks, but they should be double. There is, therefore, a lack of coherence in presentation, and suggests that it has been perhaps written by multiple people, but di video into different sections.
Firstly, I think that there are mistakes being made in presuming that readers will automatically either know or understand about the Sámi people. The explanation should be at the very beginning stating who they are exactly, what their original are. This comes too late on in the article. From the very beginning it is presumed that the reader has knowledge. Explanation is needed, for example of Fosen Vind AS. from the prologue, definitions of who the Sámi people are needs to be provided.
At the end of the prologue, there is a footnote or endnote number (i) - but there are no endnotes at all. This is rather strange for an academic article.
The author(s) state 'according to the lawyer' - but who was that lawyer and what are the sources for this?
There are no line numbers, so I have difficulty in pointing out the areas that need improvement.
In the introduction it is stated 'the colonisation process in Norway ended in 1980' - this needs to be clearly defined and explained since to a reader that knows nothing of this colonisation process, it is not understandable.
The Sámi parliament is defined as a "democratic structure representing" - how does it interact and what are its roles within the Norwegian state?
ILO - do not use abbreviations without explaining them entirely in full the first time they are mentioned.
Sápmi/Saepmi needs to be defined for all readers.
Minority within a minority - lacks in statistical data to prove this.
"we argue we need to take collective responsibility" - who is this 'we'? Not academic and should be reworded more neutrally.
"In Norway we often distinguish" - again who is this 'we'? Appears to be a general idea and opinion but not academic.
the "harsh colonisation" - this may well be true, but it is in need of academic backing and proof.
p. 3: 'we define' colonisation - again - who is this 'we'.
Kohn cited in Minde - not academically acceptable to cite someone in another author's work - cite the source from the origin.
Skille is cited directly with a quote - but there is no page number.
"employing around 150 man-years" )- can you employ man-years?
"substantially more South Sámi are involved in the industry" - where is the proof of this statement?
"court decisions since the 19th century" - precision - date? source? reference?
p.3-4 "reveal and communicate our human experiences" - needs explanation as to how and why?
"At the time of the interview" (singular?) - when? Where? circumstances - comes too late afterwards - must be here.
"living in the middle of Norway" - precision.
explanation needs to be provided as to why the transcripts were done and why verbatim?
Why choose 'he/him for all - and not 'they'? Even for females?
Why choose family members to be 'she/her'? It needs to be justified academically. Is this just a choice - apart from the obvious of anonymity - but there are many ways of anonymising people in interviews.
Were the interviews recorded? How, and what circumstances? Duration? Means?
No mention is made of oral history (Portelli Thompson, Thomson...). No mention is made of the ethics of anonymity in oral history interviews? Or, bottom-up history telling?
"we agreed that hopes and worries related to racism, language and reindeer herding" and "find quotes that would be most suitable"...this is not really the way that academic analysis should be carried out - the interviews should be done, then analysis takes place, and then they are re-grouped into fields or themes. But, the author(s) has-ve c created the themes, and then found suitable quotes.
p. 4 Indigenous methodologies, "see an analysis of this, e.g. by [27]" - explain in detail and give actual name.
"we believe that it is important" - who is 'we', again, and lacks in backing to prove what is being said.
p. 5. explanation about anonymity comes too late.
"we early on found that sex/gender was not an important analytical dimension". I doubt that is the case - but this needs to be proved why it is not important as a dimension.
p. 5 "hoped that racism would end" - racism has not been defined and should have been.
there is also a reference to language - but no backing is provided.
No reference to culture, nation-building - belonging - community - Rogers Brubaker, Benedict Anderson, Charles Taylor...
Even if the interviewees have been anonymised, they still need to be traceable by the authors - "one participant" is not sufficient (number, code should be sed in endnotes to assist the author(s) in relocating the interview at a later stage.
"if you were engaged in reindeer husbandry" - who is the 'you'?
What language were these interviews done in?
"the incident happened on a warm sundry day" - is this relevant?
"But how wrong he was!" - not academic in writing style...judgemental a nd emotional, rather than scientific. Author(s) need to be more removed and stand back.
"hopes and worries about the South Sámi language" - is this a subtitle?
325% estimated to speak the South Sámi language" - proof, source, reference?
"he said giggling but worried" - emotion in oral history interviews should be cited, and needs greater analysis.
The quotes lack somewhat in analysis.
p. 8 "most prevalent among Sámi practising reindeer herding as a lifestyle" - analysis is needed.
"re-establish justice" - what does this mean, exactly. Seems less academic and more political. It needs to be justified.
Discussion has some good references, but sometimes the points are being repeated from the previous section.
"The truth and Reconciliation Commission could potentially play an important role" - why how so? Explain.
"Alta dam" - needs explanation for those who do not know.
"we may hope that justice" - who is this 'we' again?
"concerns all of us" - according to whom and why - explain and justify with academic backing.
In acknowledgements it states (Remember PM, Canada) - what does this mean?
Author Response
Dear reviewer. Thank you so much for your review. Please see attached file and the new manuscript draft, where you can see our changes in red.

Reviewer 2 Report
Comments and Suggestions for Authors
This is an excellent article - clear, focused, and well-documented! It provides a fascinating description and analysis of the South Sami and their struggles with colonialization.
Author Response
Dear reviewer. Thank you so much for your review, attached you will find our response and our revised version of the article draft. We hope this is sufficient.

Reviewer 3 Report
Comments and Suggestions for Authors
Argument: Even though colonization of the South Sami by Norway ended in the 1980s, the Sami still experience the effects of colonization, including the threat of displacement from their ancestral lands by the development of wind parks. Elderly South Sami are cautiously positive about the future but worry about whether their children and grandchildren will face anti-Sami racism and ongoing threats of losing their herding land to developmental projects.
The author’s conclusions feel extremely slight in terms of its significance. I find myself asking:
(1) How does this differ from how South Sami elders felt a few decades ago?
(2) How does this compare with how this came age cohort felt about the same issues in earlier life stages?
(3) How does this compare with how other age cohorts of South Sami feel about these same issues?
(4) How does this compare to how elders in other Sami communities feel?
What I am trying to get at here is that results gain significance when they tell us something about what has changed or not changed with relation to a specific situation. For example, the article’s authors mention the 1980s as a turning point in Norway’s relations with its Sami communities. Is the authors’ main purpose to understand how the changes in the 1980s have affected South Sami people’s outlook on their future? If so, the authors need to say more explicitly what the 1980s Truth and Reconciliation set out to do for South Sami communities and whether, based upon the participants’ responses, those goals have been met or whether those goals were even appropriate and effective ways to resolve the issues that the South Sami elders themselves identify as most threatening to their communities’ survival. This, in turn, can lead the authors to a broader, more significant conclusion, like: (
1) The 1980s reforms have/have not changed South Sami elders’ outlook on their community’s future
(2) There has been/has not been a fundamental disconnect between what the Truth and Reconciliation process sought for the Sami and what the Sami themselves wanted or needed.
Methodology:
I have a small quibble here and then, a much larger issue.
Small quibble:
What is a “partly Indigenous” person? A person who happens to be genetically less than 100% Indigenous, but might still be closely connected to their Indigenous community? A person who has living Indigenous family members, but is not personally involved in their community and culture for reasons of adoption, residential school, or other colonial-institutional issues that continue to separate Indigenous children from their communities? A person who has one Indigenous ancestor 10 generations ago? A person who self-identifies as Indigenous, but has no living connection to an Indigenous community? (To better understand what I am getting at here, please see the work of Kim TallBear.)
“Partly Indigenous” is a category that might make sense to settler readers, but will read strangely to Indigenous readers. Also, the authors’ current division—“Indigenous,” “partly Indigenous,” “allies”—presents (probably unintentionally) “Indigenous” as a monolithic category. A Dakota person researching Dakota issues among Dakota people is in a very different place from a Dakota person doing research about Nuche issues in a Nuche community, to say nothing of a Dakota person researching in a Sami community.
What you really mean to say is that it is good practice for research in Indigenous communities to be conducted by researchers who come from within the communities with whom they are working.
Go ahead and drop the “partly Indigenous,” a term that really isn’t helpful anyway, and focus on the Indigenous community members researching in their own communities and the outsider ally researchers.
Now for the larger methodological issue:
I understand the authors’ desire to preserve the anonymity of their participants, but by withholding all biographical specifics about these people including their gender, the authors make it hard to draw much in the way of conclusions. For example, did being male or female make a difference in the ways that the South Sami elders experienced racism? Did living in an urban area for most of one’s life versus living in a rural area make a difference? Did being raised herding reindeer make a difference in the informants’ opinions? How about being fluent in their language versus not knowing it? Is there any difference in perspective between those elders who may have been politically active in the service of Sami status and decolonization versus those who have not been?
What I am trying to get at here is that there is a lot that could be said about how life experience, upbringing, socio-economic standing, gender, and political activism affect the way that South Sami elders think about the future of their community. All of that is erased when the authors insist on dealing with all their informants as generically Sami, generically old, and nothing else.
I do not know what the authors contractually promised to their informants, and I realize that they may be circumscribed by contractual obligations. However, one of the major injustices committed by non-Indigenous researchers working in Indigenous communities is the erasure of Indigenous people’s individuality and the reduction of Indigenous individuals to a single racial-cultural type. This reduction of Indigenous people to a single type is juxtaposed to the tendency to view settlers as unique individuals. However good the authors’ intention, they have taken their adherence to anonymity to such an extent that they risk perpetuating this colonial tendency. Sami elders are not all the same person. Each one of them has a personality and life experiences and relatives and jobs and politics and opinions. I would urge the authors to think of ways to bring this diversity across even while working within their framework of anonymity.
Finally, as the authors have expressed their interest in conducting ethical research, I would ask them: What does the South Sami community stand to benefit from your research? What are you doing to make sure that you are not just another team of settlers coming into the community to exploit it for your own career ambitions? Are you providing the results of your research to the community? Is your research being used by the community to further its struggle for cultural and linguistic preservation/revival in a post-colonial situation? Given the methodological framework and ethical goals you have set out for yourselves, you really should provide an answer to this question in your article.
Author Response
Dear reviewer. Thank you so much for your review. Attached you will find our response. You will find our changes in red in the new manuscript draft. All the best.
(From the editorial office: Please scroll down towards the end of the first page of the author response file for the authors' response to your initial comments.)

Round 2
Reviewer 1 Report
Comments and Suggestions for Authors
This is a vast improvement on the previous submission.
The authors have improved the scholarly references and in particular regarding ethical considerations, racism and the use of language, and the context of colonisation.
This is good.
There are still no line numbers, however.
On page 4, it is really unacceptable in terms of scholarly analysis to cite an author that is citing another author. The original author must be cited and not someone who cites within a citation. It is not academically acceptable. "MN Sara, 2009 in Johnson, 2018". Cite Sara. I did state this previously.
There are occasions when there is too much repetition of the same elements:
page 3 paragraph 2 "state-imposed colonisation officially ended decades ago". This is repeated throughout. Change the way it's said.
Similarly, page 11: "Sami" - accent is missing. "South Sámi language by taking classes if they could not already speak the South Sámi language" - and other places in this paragraph. replace with it, don't keep repeating SSL.
At the bottom of page 11, the authors provide a quote. But the quote does not bring anything more than what is said in the preceding paragraph. Change what is said before and use different words. The authors do not say how learning that language or taking classes would happen. Have there been attempts? Are there schools? Is there an increase in the numbers taking classes? Are there teachers?
page 3, the authors mention "from a population over approximately 80-95,000" - source? When? What date was the population at this level?
"especially those who represent the majority population" - what does this mean? Do you mean the government? Policy-makers?
Page 5 - and then throughout the paper Skille quotation - double quotes should be used, not single for quotations.
Page 7 - you mention life stories - but you mention none of the references I previously suggested about the importance of bottom-up life-story telling and oral history. This should be incorporated. (Portelli, Thompson, Thomson).
Page 7 and 8 - on page 8 the authors speak of anonymity. But, what was said on page 7 about this is too far removed and should be joined together.
page 9 - I did state before that the fact that gender is not important in the eyes of the authors must be explained why they believe that it has little or no importance. Women are always subjugated to greater effects in the course of discrimination. If they believe it not to be the case, then they must defend and explain to prove that.
page 17 - there are no proposals made. What do the Sámi want? Programmes? Teaching? Classes? Information? Suggest what should be done to improve the situation.
The authors speak of healthcare : so what is the proposal to improve the situation of knowledge of Sámi people?
The bibliography is problematic:
Some elements are in bold. To be changed.
Number 22, for example KOHN has no first name. The authors also use initials for all works. The full name should be used?
The Internet sites do not have the 'date of consultation" listed after them every time.
The format of the works is incorrect and does not correspond to norms. Number 43, for example - but they are all the same - the title of the article should be in single inverted commas, and the title of the academic journal should be in italics.
Comments on the Quality of English Language
The punctuation needs to be checked. For example, page 2 "In 1968" - there should be a comma here. Check throughout, as there are other times, also.
page 2 "all in the society has" - all in society have.
page 5 "Skille [24] also display that government" - also displays.
"In the Sámi-Norwegian history"
sub-ordination - replace with "subordination".
page 6 - "the aim was" - replace with "is".
"whereas other" - others
page 7 "as XX do not herself" - incoherent verb
page 10 "one of them, have give this example when talking about the coming generation said:" - comma incorrect and said is incompatible with what precedes.
"experienced racism" - have experienced.
page 14 - "in regard of" - with regard to
"experienced racism today" - have experienced
page 16 - "led to that the South Sámi language is severe threatened" - led to the fact that the SSL is severely threatened.
"existed in generations" - over generations?
Reviewer 3 Report
Comments and Suggestions for Authors
The authors have addressed the issues that I raised in my initial review.
Methodology:
The authors do a better job in this new draft of defining their own relationship to the community they are working with. They also addressed the concerns I raised about defining "indigenous" vs. "partly indigenous" researchers.
Analysis of Data:
The authors addressed the concerns that I raised about over-anonymizing their informants by giving more information on the community in general and the specific categories of people within the community they were focusing on for this article.
The authors also devoted more space in this version of the article to putting their informants' concerns in dialogue with and in the context of recent decolonization and Truth and Reconciliation processes. This gives readers a better idea of the successes and failures of these processes as viewed through the eyes of the Sami informants. It also highlights potential flaws in these processes.
Author Response
Thank you so much for this. We really appreciate the effort.